# Flexural Behavior of Damaged Hollow RC Box Girders Repaired with Prestressed CFRP

**DOI:** 10.3390/ma16093338

**Published:** 2023-04-24

**Authors:** Xinyan Guo, Lingkai Zeng, Xiaohong Zheng, Baojun Li, Zhiheng Deng

**Affiliations:** 1School of Civil Engineering and Transportation, South China University of Technology, Guangzhou 510640, China; xyguo@scut.edu.cn (X.G.); ctlkzeng@mail.scut.edu.cn (L.Z.); 2Guangxi Transportation Science and Technology Group Co., Ltd., Nanning 530001, China; jamberly@163.com; 3College of Civil Engineering and Architecture, Guangxi University, Nanning 530004, China; dengzh@gxu.edu.cn

**Keywords:** prestress, carbon fiber-reinforced polymer (CFRP), hollow RC box girders, damage, FEA

## Abstract

In recent years, numerous studies have explored the benefits of utilizing prestressed carbon fiber-reinforced polymer (CFRP) for strengthening concrete structures. However, research on the reinforcement of prestressed CFRP on full-scale hollow RC box girders, particularly damaged bridges, remains limited. In this study, both experiments and finite element analysis (FEA) were performed to investigate the flexural behavior of full-scale hollow RC box girders with varying degrees of damage, which were strengthened using CFRP with different levels of prestress. The adhesive behavior of the CFRP–concrete interface was considered in the FEA. Numerical simulations were conducted to assess the flexural behaviors of the girders, including failure modes, yield and ultimate loads, and deflections. The results revealed that the application of prestressed CFRP efficiently increased the yield and ultimate loads of the box girders. Specifically, when the degree of damage of the hollow box girder was less than 23%, the flexural bearing capacity of the repaired girder could be recovered after being strengthened with two prestressed CFRP strips measuring 50 mm in width and 3 mm in thickness. However, the risk of premature debonding at the CFRP–concrete interface increased when the prestressing level of CFRP and degree of damage of hollow RC box girders exceeded 35% and 40%, respectively. These findings suggest that the use of prestressed CFRP may be a promising method for repairing damaged hollow RC box girders, but careful consideration of the degree of damage and prestressing level would be necessary to ensure the effectiveness and safety of the repair.

## 1. Introduction

In recent years, the performance of many bridge structures has significantly degraded due to various factors, including corrosion, heavy traffic, deicing salts, vehicle impact, and other adverse conditions. Strengthening has become an effective approach to maintain deficient bridge structures in a safe operating mode. Currently, the use of fiber-reinforced polymers (FRPs) for strengthening has gained widespread attention owing to their high tensile strength, excellent corrosion resistance, and lightweight properties. 

In early studies, undamaged reinforced concrete (RC) girders were used for researching fiber-reinforced polymer (FRP) strengthening techniques. However, many bridges and other RC structures requiring strengthening were already damaged. To address this issue, some researchers conducted tests to investigate the effectiveness of FRP strengthening on damaged RC members. Thanoon et al. [1] investigated the flexural behavior of cracked RC girders repaired using five different techniques. They found that carbon fiber-reinforced polymer (CFRP) strip reinforcement was superior to other reinforcement technologies. Benjeddou [2] conducted tests to investigate the behavior of damaged RC girders repaired with CFRP. Four different degrees of damage (0%, 80%, 90%, and 100%) were preloaded, and the damaged girders were then repaired using CFRP strips with a width of 100 mm. The results indicated that CFRP significantly improved the load capacity of the repaired beams, regardless of their degree of damage. After strengthening, the load capacity and rigidity of the RC beam with 100% damage were higher than those of the undamaged control beam. Siad [3] produced pre-cracked RC beams by preloading them up to 60% of the ultimate load and exposing them to an environment with three different accelerated corrosion rates. After being strengthened with CFRP strips with a width of 100 mm, the ultimate strengths of the strengthened beams were higher than those of the control beams without pre-cracking or corrosion. Xie [4] evaluated the effectiveness of applying CFRP to repair corroded reinforced concrete beams using an improved method based on substrate repair. The results showed that fully bonded CFRP can provide better load-carrying capacity for corrosion-damaged RC beams. Wang [5] and Masoud [6] investigated the damage mechanism of test beams reinforced with CFRP for corrosion damage. They concluded that strengthening corroded bridges with FRP not only improves the load-carrying capacity of the beams but also increases their resistance to further erosion.

Directly bonding CFRP externally cannot fully utilize the high strength of CFRP. Therefore, prestressing technology, as an active reinforcement method, has gained widespread attention in the field of CFRP-reinforced bridges [7,8,9]. The advantage of prestressing technology is that it can generate negative bending moments to increase the load capacity of the bending member. Thus, it effectively improves the reinforcement’s effect. Several scholars demonstrated that prestressed CFRP reinforcement can make full use of the tensile properties of CFRP materials more than direct CFRP reinforcement [10,11]. 

To repair damaged full-size bridges, non-prestressed and prestressed CFRP strengthening methods are used. Gardern [12] reinforced two 18.0 m long damaged beams from a failing bridge using 30% of the ultimate strength of prestressed CFRP. The load-carrying capacity of the strengthened beams increased by up to 60%. Andrak et al. [13] repaired Lauter Bridge, a metal bridge with two spans of 7.8 m, using a prestressed CFRP laminate with 1000 MPa stress. The ultimate load increased from 7.5 to 40 t by using the CFRP prestressing technique. Kim [14] repaired a prestressed concrete girder bridge with a C-shaped cross-section, which was damaged by impact loads, using prestressed CFRP sheets, with a prestress level of up to 21% of the ultimate fiber strain. The damaged bridge’s flexural behavior was recovered to an undamaged state after repair. John Aidoo [15] reinforced eight T-shape bridge girders from a decommissioned bridge using three different CFRP methods, conventional adhesive application (CAA), near-surface mounted (NSM), and powder-actuated fasteners (PAF), and it was observed that the behavior of the bridge with the lower grade design increased after CFRP reinforcement.

In addition to experimental studies, several theoretical analyses have also been conducted [16,17,18,19,20,21]. Kabir [16] used an elastic assumption to calculate the load capacity of CFRP-reinforced structures, considering the anchorage effect at the end of the CFRP. Azimi and Sennah [17] used finite-element analysis (FEA) and load resistance methods to calculate the live load distribution factor of a damaged bridge girder repaired with CFRP. Kasan et al. [18] conducted a parametric study on the strengthening of damaged prestressed concrete girders using two shapes (box and I-girder) and three CFRP repair methods: externally bonded (EB), bonded post-tensioned (bPT), and near-surface mounted (NSM). They found that the specimen should have enough residual capacity before repair, and the required area was important when selecting the repair method. Many scholars have used a combination of experimental and finite element simulations to analyze the performance of FRPs [19,22,23,24,25]. Jin et al. [19] developed a numerical model using the FEA method to analyze the mechanical behavior of damaged concrete RC girders reinforced with CFRP. They discussed the influence of the U-shaped sheet and the number of strengthening layers and recommended the use of four layers of CFRP and U-strips.

From the mentioned research, the advantage of using prestressed CFRP for strengthening damaged bridge structures is obvious. However, because the box girders were hollow inside, the use of prestressed CFRP in strengthening full-scale hollow box girders is still limited. In China, 25% of bridges are made of hollow box girders, making it necessary to develop effective methods for their strengthening. However, numerical simulations that consider the behavior of the interface between CFRP and concrete are seldom involved in the analysis of RC beams strengthened with prestressed CFRP due to their complexity. In this study, experiments and numerical analysis of a damaged hollow box beam strengthened with prestressed CFRP were carried out. The flexural behavior, including failure mode, load capacities, and shear stress distribution of the CFRP–concrete interface, was investigated based on the numerical model. Furthermore, the influence of prestress level and degree of damage on the flexural behavior of the specimen was discussed.

## 2. Experimental Program

Two decades ago, hollow box girders accounted for 25% of the total number of bridges in China due to their cost efficiency and installation convenience. However, due to overloading, heavy traffic, critical environments, and design limitations, the performances of some bridge structures have significantly degraded. According to bridge management information, by the end of 2016, 44% of hollow box girders had structural faults. In this study, four damaged full-scale box girders were tested, with one serving as a control girder without strengthening and the other three having been strengthened with prestressed CFRP. Quasi-static loading was used to test the flexural behavior of the box girders after strengthening.

### 2.1. Typical Damage to Concrete Hollow Box Girders

Typical damage to hollow box girders includes shear cracks, flexural damage, parallel bending cracks, roof damage, and steel bar corrosion, as shown in Figure 1.

*Shear cracks*. During the design process, the flexural bearing capacity of the hollow box girders was the main focus. Unfortunately, the shear bearing capacity was neglected, and the length of the stirrups was designed to be short, whereas the web was very thin. As a result, some shear cracks appeared at the end of the girder over the service period. 

*Flexural damage*. Hinged joints were utilized to transfer loads between the hollow box girders. When the hinged joints were damaged, the load transfer capability was significantly weakened, and forces were then concentrated on a single girder. Consequently, flexural damages such as concrete cracks at midspan, significant deflection, concrete spalling, exposure of main reinforcement, and concrete ruptures were observed.

*Parallel bending cracks*. Due to increasing traffic and overloading, the designed flexural capacity was less than the actual service load. After several years of operation, parallel bending cracks appeared at the bottom, resulting in a reduced effective concrete height and significantly impacting flexural capacity. The distance between the appearing bending cracks was approximately 15–20 cm.

*Roof damage*. During the precast process, inflatable rubber cores were placed inside the hollow girders, which made them prone to float and resulted in a lower roof thickness than the designed thickness. Furthermore, the roof was damaged during transportation to the construction site and was not repaired promptly. During service, the impact of vehicles can also damage the roof of the hollow plate.

*Steel bar corrosion.* In northern regions, snow-melting agents are commonly used during winter to remove snow. However, the presence of salt in the agent accelerates the corrosion of steel bars in concrete, which ultimately results in concrete spalling and swelling cracks. Furthermore, this corrosion process is further accelerated, leading to a significant decrease in the structural integrity of the concrete.

### 2.2. Description of Actually Damaged Hollow RC Box Girders

In the current study, four full-scale damaged hollow RC box girders were obtained from an old bridge constructed in China in 2002. The cross-section of the old bridge is illustrated in Figure 2. After two decades of service, some visible cracks appeared in the bridge girders. The full-scale hollow box girder has dimensions of 15,960 mm in length, 750 mm in height, and top and bottom girder widths of 840 mm and 1040 mm, respectively. The main reinforcing steel rebars at the bottom of the box were 10Φ25 mm, whereas it was 2Φ22 mm at the top girder. The diameter of the stirrups was 8 mm, and the concrete cover thickness was 45 mm. The details of the cross-section of the damaged RC box girder are also presented in Figure 2.

According to the specifications for inspection and evaluation of the load-bearing capacity of highway bridges [26], the overall technical conditions of the bridge are classified into five categories based on the technical condition evaluation criteria: functional integrity, mild damage, moderate damage, severe damage, and serious damage. It is possible to evaluate the damage level *D* of the hollow box girder in real conditions. After the inspection, the damage levels of four girders, G1, G2, G3, and G4, were found to be 7%, 31%, 35%, and 10%, respectively.

### 2.3. Prestressing Applied

According to the manufacturers, the width and thickness of the CFRP are 50 mm and 3 mm, respectively. Given the thinness of the concrete beneath the hollow box, drilling holes to install the anchorage system within this range poses a safety hazard. As a result, the location for prestressed CFRP was determined to be 350 mm from the girder axis. Figure 3 illustrates the schematic for prestressing the CFRP.

The hollow box girder with minor damage without strengthening was denoted by G1, and the other three damaged box girders (G2, G3, and G4) were strengthened by CFRP with 30%, 40%, and 60% prestressing levels, respectively. The details of the specimens are listed in Table 1.

The reinforcement processes included the following: (i) the bottom surface of the damaged RC girder was repaired with primer and putty; (ii) the anchors were bolted to the bottom of the damaged RC girder end, as shown in Figure 4a; (iii) the epoxy resin was applied to the surface of the CFRP and concrete; (iv) the CFRP was installed on the anchor system; (v) the prestress was applied on the CFRP with external force; (vi) to reduce the debonding failure on the CFRP–concrete interface, four mechanical fasteners were installed on each CFRP, as shown in Figure 4b.

### 2.4. Load Experiment

As shown in Figure 5, the damaged RC box girders were subjected to four-point bending across a span of 15.36 m. The load was applied under field conditions using hydraulic jacks that acted on rigid counterforce frames composed of steel girders and large stones. Moreover, the loading value was measured using a static data acquisition instrument connected to the loading sensor. The views of the field conditions and loading device are also presented in Figure 6 and Figure 7, respectively.

Fourteen displacement gauges were symmetrically placed at seven control points to measure the deflections of the specimens, including two supports, two quarter-spans, two loading points, and the mid-span. Each control point was equipped with two displacement gauges.

The bending capacity of the four girders is listed in Table 1. In previous studies [27], relevant parameters of the damaged hollow box girders were measured, and the degree of damage to the girders were calculated.

## 3. FE Model

To improve the efficiency of the strengthening process, FE models were developed to predict the flexural behavior of damaged RC box girders strengthened with different levels of prestressed CFRP. The FE model of the damaged RC box girder was built based on the specimen’s details. Using the finite element analysis software ABAQUS, the flexural behavior of the damaged hollow box girders without strengthening and after strengthening with prestressed CFRP was analyzed.

By taking advantage of symmetry to reduce computational time, the detailed arrangement of the quarter hollow box girder was generated and illustrated in Figure 8, which shows the schematic of the prestressed CFRP location and the model of the prestressed CFRP-strengthened girders.

### 3.1. Material Properties 

For the concrete, 3D-integrated hexahedral elements with eight nodes (C3D8R) were used for simulation. The stress–strain relationship of concrete is a non-linear model provided by the standard GB50010-2010 [28]. The tensile strength, compressive strength, elastic modulus, and Poisson’s ratio of the concrete were *f*_t_ = 2.65 MPa, *f*_c_ = 32.4 MPa, *E*_c_ = 35.0 GPa, and *v*_c_ = 0.20, respectively. Additionally, the plastic damage model (CDP) was used to simulate the behavior of concrete, considering the concrete model provided in the software. The parameters of tensile and compressive damage were also imported into the FE model. The uniaxial stress–strain relationship of concrete is presented in Figure 9a.

The main reinforcing steel bars, shear reinforcing steel bars, and stirrups were all made of typical HRB335 with different diameters and were established with 2D truss elements with three nodes (T3D2). The stress–strain relationship of reinforcing steel was modeled as an elastoplastic model, as shown in Figure 9b. The yield strength, elastic modulus, and Poisson’s ratio of the reinforcements were *f*_y_ = 355 MPa, *E*_s_ = 206 GPa, and *v*_s_ = 0.3, respectively. Moreover, in the present study, the connection between the reinforcing steel and concrete was assumed not to slip and was simulated with embedded constraints in the FE model.

CFRP is a material known for its high strength, light weight, and non-corrosive properties. Nowadays, CFRP is widely used as a prestressing material for bridge strengthening. In this study, a four-node, doubly curved thin or thick shell (S4R) was used for its accuracy in bending simulations. Moreover, the behavior of CFRP was modeled using the linear elastic model. The tensile strength, elastic modulus, and Poisson’s ratio of CFRP were *f*_pu_ = 2400 MPa, *E*_p_ = 16.1 GPa, and ν_p_ = 0.25, respectively. 

In the model, the approach for modeling prestressing levels of CFRP involved applying initial stress. The prestressing level was defined as the percentage of the ultimate tensile strength of CFRP, which is 2400 MPa.

### 3.2. CFRP–Concrete Interface

Previous studies [29,30,31,32,33] show that CFRP–concrete interface debonding is one of the four main damage modes of RC girders when strengthened with FRP. High prestressing levels of CFRP used for strengthening RC girders may lead to a higher likelihood of interface debonding. Therefore, a crucial element for the accuracy of the FE model is the bond–slip relationship at the CFRP–concrete interface, as it plays a crucial role in the load transfer from the hollow RC box girder to the CFRP. Researchers found [34] that the bond–slip relationship obtained from single and double shear tests is not suitable for RC girders due to cracks at the bottom of the concrete, which can cause rapid performance deterioration. Hence, the bond–slip model of the bending specimen used in the FE model was the simplified bond–slip relationship modeled in Equation (1) and shown in Figure 10,
(1)$$\tau =\{\begin{array}{cc} \tau_{max}\sqrt{\frac{S}{S_{0}}} & S\leq S_{0}\\ 0 & S>S_{0} \end{array}$$
where $\tau_{max}$ is the CFRP–concrete interfacial bond strength and *S*_0_ is the slip between CFRP and concrete corresponding to the interfacial bond strength. $\tau_{max}$ and *S*_0_ can be obtained with the following equations [34]:(2)$$\tau_{\max}=1.5\sqrt{\frac{2.25-b_{f}/b_{c}}{1.25+b_{f}/b_{c}}}f_{t}$$
(3)$$S_{0}=0.0195\sqrt{\frac{2.25-b_{f}/b_{c}}{1.25+b_{f}/b_{c}}}f_{t}$$
where $f_{t}$ is the tensile strength of concrete and $b_{f}$ and $b_{c}$ are the width of the CFRP and concrete specimens, respectively.

In this study, the bond–slip model shown in Figure 10 was adopted, which has been widely used and validated in previous studies. The accuracy of the bond–slip relationship considered in the FE model has been demonstrated in many studies. Based on test results [35], the main bonding parameters of the epoxy adhesive were $\tau_{\max}=5.301\mathrm{MPa}$ and $S_{0}=0.0689\mathrm{mm}$, which were in good agreement with the values calculated using Equations (1)–(3). The debonding failure occurred when the greatest shear stress on the CFRP–concrete interface reached the interfacial bond strength $\tau_{\max}$.

### 3.3. Failure Criteria

Considering that the main residual capacity of RC girders is affected by the main reinforcement, the degree of damage was simulated by reducing the corresponding cross-sectional area of the main reinforcement steel bars.

For RC girders strengthened with prestressed CFRP, the main failure modes include CFRP fracture at the bottom of the midspan, concrete crushing at the top of the RC girders, yielding of the main reinforcement steel bars, and CFRP–concrete interfacial debonding failure [7]. Many studies [7,27,35] have shown that CFRP–concrete interfacial debonding failure and CFRP fracture frequently occur with increasing prestress levels. So far, most concrete specimens used for prestressed CFRP strengthening experiments have had rectangular or T-shaped cross-sections, and there have been few studies describing the failure modes of hollow box girders strengthened with prestressed CFRP. The FE analysis included four failure criteria: CFRP fracture, CFRP–concrete interfacial debonding failure, concrete crushing, and yielding of the main reinforcement steel bars. The failure criteria were defined as follows based on the material descriptions: (i) concrete reaches ultimate compressive strain *ε*_cu_ = 0.0033; (ii) steel reaches yield strength σ_s_ = 335 MPa; (iii) interfacial shear stress reaches $\tau_{\max}=5.301\mathrm{MPa}$; (iv) carbon fiber reaches tensile strength σ_pu_ = 2400 MPa. During the loading process, if any of the failure criteria were met, the corresponding failure of the bridge occurred.

### 3.4. Damage Setting

Defining the extent of damage to the decommissioned bridges studied is crucial for finite element simulation. Based on previous research [27], the damage coefficients for girders G1–G4 were calculated and are shown in Table 1. To simplify calculations and analysis, the cross-sectional area of the steel bars was reduced in the finite element model to simulate the girder’s damage. The specific formula used is as follows:*A* = (1 − 0.1*D*)*A*_0_(4)
where *A* is the reduced cross-sectional area of the bar, *A*_0_ is the initial area of the bar, and *D* is the damage coefficient.

### 3.5. The Sensitivity Analysis of Mesh Size in FE Modeling

In the finite element analysis, three different mesh sizes of concrete were chosen to explore mesh sensitivity and the convergence of the different mesh sizes, as shown in Table 2.

It can be seen from Table 2 that when the mesh size is less than 100 mm, the influence of the mesh size on the analysis results is relatively small, whereas when the mesh size is 200 mm, the yield load and ultimate load increase significantly. Therefore, considering analysis accuracy and calculation efficiency, the concrete mesh size of 100 mm was chosen for all models in this paper.

The reinforcement cage, on the other hand, was analyzed with a mesh size of 400 mm, taking into account the overlap between the reinforcement bars, whereas the 50 mm transverse and 100 mm longitudinal mesh were chosen for the CFRP analysis. After finite element simulations, the models all converged.

### 3.6. Comparison of FE and Test Results

This section presents the validation of the FE model by comparing the load–displacement curves produced by the experiment and finite element simulation. The comparison of the load–displacement curves of the four hollow RC box girders is illustrated in Figure 11 and Table 2. As shown in Figure 11, the load–displacement curves can be divided into three stages. The first stage is the linear elastic stage. The cracking load in the experiment of load–displacement curves was not visible for the crack existing before strengthening. However, in the FE numerical analysis, the damage level was simulated by reducing the cross-sectional area of the steel reinforcement, and concrete cracking was evident when the load reached the cracking load.

After the first stage, the second stage is the load increasing stage. During this stage, the height and width of concrete cracks increased. However, when the load reached the inflection point P_1_, the longitudinal tensile reinforcement in the hollow RC box girder began to yield. The last stage is the final stage to failure, and the ultimate strength of the specimens is denoted by P_2_.

The load–deflection curves obtained from the finite element simulations show good agreement with the actual experimental conditions in terms of overall trends. As shown in Figure 11a, the maximum error between the finite element and experimental curves was 10.2%, and the yield and ultimate strength of the experimental and numerical results are almost in complete agreement. The error data for a few specific points are presented in Table 3. It can be seen from Figure 11a that the load–deflection curve obtained from the experiment was slightly higher than that of the finite element analysis, which is possibly because the experimental girder was partially crack-patched with epoxy resin adhesive before the experiment was carried out.

## 4. Analysis of Influencing Factors

This section explains the investigation of damage degree and prestressing level’s effects on the mechanical behavior of prestressed CFRP-reinforced hollow box girders using the FE model. To facilitate analysis and comparison, the FE models were labeled with parameters such as damage and prestressing level. For instance, the FE model D5-P30 represents a hollow box girder with a degree of damage of 5% and a CFRP prestressing level of 30%.

### 4.1. Effect of Degree of Damage

According to the Specification for Inspection and Evaluation of Load-Bearing Capacity of Highway Bridges [24], the technical condition of a bridge is classified into five categories: functional integrity, mild, moderate, severe, and serious damage. This is based on the technical condition evaluation criteria. This paper focuses on the finite element simulations of prestressed CFRP-reinforced hollow RC box girders with minor, moderate, and major damage (corresponding to 5%, 20%, and 35% degrees of damage, respectively) to investigate the effects of degree of damage on the bearing capacity, reinforcement strain, and CFRP–concrete interface of damaged hollow RC box girders.

#### 4.1.1. Load–Strain Curves of Main Reinforcement

The load–strain curves of the longitudinal reinforcement of the prestressed CFRP-reinforced hollow box girders at various degrees of damage are depicted in Figure 12. As observed in Figure 12, the strain of the main reinforcement remained almost constant before cracking and increased slowly with the increasing load. After the cracking load, the slope of the strain curve of the main reinforcement increased significantly. When the load reached the yield load, the longitudinal reinforcement yielded, and the strain of the reinforcement increased rapidly with the increasing load until the girder failed. The load–strain curve also shows that there was a significant difference in the load at which the reinforcement yielded for different degrees of damage, with larger degrees of damage resulting in smaller loads that the reinforcement could bear before it reached yield. Furthermore, as the degree of damage increased, the reinforcement reached yield earlier.

#### 4.1.2. Strength

The yields and ultimate loads of RC girders with different degrees of damage, strengthened with CFRP with a 30% prestressing level, are presented in Figure 13. It can be observed that the yields and ultimate loads of the girders with greater degrees of damage were smaller than those of the girders with lesser degrees of damage for a certain prestressing level. For instance, the yield load of D35-P30 with 35% degree of damage was 70.52 kN. When compared to the models (D5-P30 and D20-P30) with 5% and 20% degrees of damage, the yield load of D35-P30 decreased by 14.10% and 11.24%, respectively. Similarly, for the ultimate load, that of D35-P30 with greater damage was 73.19 kN. The ultimate loads of the RC girders (D5-P30 and D20-P30) with 5% and 20% degrees of damage decreased by 12.97% and 6.55%, respectively. The strengthening effect of the slightly damaged RC hollow box girders was significantly better than that of moderately damaged and highly damaged girders. Moreover, the greater the degree of damage, the worse the effect of the prestressed CFRP-reinforced RC hollow box girder. As the degree of damage increased beyond 20%, the strengthening effect decreased more significantly.

Based on the aforementioned data, it is imperative to reinforce RC hollow box girders in service when their degree of damage reaches 20% during health inspections, as the reinforcement effect will be significantly reduced as the damage progresses.

Upon reaching a degree of damage of 23%, the ultimate bearing capacity of the RC reinforced girder can be restored to 520.36 kN by applying a 30% prestressing level of CFRP, which is identical to the ultimate load of 520 kN in the initial undamaged state. However, if the degree of damage exceeds 23%, the reinforcement effect of the 30% prestressed CFRP (50 mm width and 3 mm thickness) will no longer suffice to fully recover the bearing capacity of the box girder.

#### 4.1.3. CFRP–Concrete Interface

Figure 14 displays the CFRP–concrete interface of specimens with varying degrees of damage below the ultimate load. The red regions represent areas where the interfacial shear stress between CFRP and concrete reached the shear strength limit. At a constant prestressing level of 30%, the CFRP–concrete damage at the anchorage end was greater and gradually expanded towards the middle of the span. This suggests that, with an increasing load, the CFRP debonds from both ends and propagates toward the middle. The likelihood of CFRP debonding increases as the degree of damage to the hollow RC box girders becomes greater.

The interfacial CFRP–concrete damage of the FE model D35-P30 when the main reinforcement yields is shown in Figure 15. The results show that when the initial degree of damage of the hollow RC box girders reached 35%, steel yielding and CFRP–concrete interface debonding occurred simultaneously. In other words, the failure mode of specimen D35-P30 was characterized by steel yielding and CFRP–concrete interface debonding. Consequently, when the degree of damage exceeded 35%, the damage mode of the CFRP-reinforced hollow RC box girder transitioned to CFRP–concrete interface debonding.

### 4.2. Effect of Prestressing Level

Increasing the prestressing level of CFRP can enhance the flexural performance of RC girders. However, higher prestressing levels do not always yield better results, as they may lead to CFRP fracture and premature debonding of the CFRP–concrete interface, thereby affecting the effectiveness of the prestressed CFRP reinforcement. In this section, moderately damaged (20% damage level) RC hollow box girders were selected, and the prestressing level ranged from 10% to 60%. We investigated the effect of CFRP prestressing levels on the flexural performance of damaged RC hollow box girders and the allowable prestressing degree.

#### 4.2.1. Strength

The yield and ultimate load values of a 5% damaged hollow RC box girder reinforced with different CFRP prestressing levels can be seen in Figure 16. It can be observed that for slightly damaged girders, the strengthening effect of box girders reinforced with prestressed CFRP is remarkably evident. The yield loads of D5-P30 and D5-P60 are 477.41 kN and 520.81 kN, respectively. When compared to the yield load of the unreinforced girder, the yield loads of the two strengthened specimens increased by 9.2% and 19.2%, respectively.

The results obtained suggest that in comparison to yield load, the effect of the ultimate load from prestressed CFRP is more significant. The ultimate loads of the D5-P30 and D5-P60 were 563.36 kN and 602.03 kN, respectively, which were 16.2% and 24.2% higher than that of the unreinforced hollow RC box girder. The finite element results obtained showed good agreement with the experimental data, indicating that when the degree of damage was mild, the effect of the hollow RC box girder strengthened by prestressed CFRP was significant. Additionally, the strengthening effect was observed to increase with higher prestress levels when the degree of damage was slight.

#### 4.2.2. Load–Deflection Curve

The load–deflection curves of hollow RC box girders with different prestressing levels at a 20% degree of damage are given in Figure 17. In the figure, the area to the right of the red dashed line indicates that the reinforcement yielded at that time, indicating that the effect of different prestressing degrees on the reinforcement effect was already reflected when the reinforcement had not yet yielded. The reinforcement effect of high prestressing was more obvious after the reinforcement had yielded. As the prestressing level increased, the yield and ultimate loads also increased. Nonetheless, when the prestressing level reached 30%, the improvement in the reinforcement effect became insignificant. High prestressing levels may not always bring good effects if the RC girder is in poor condition with a high degree of damage. Based on the previous sections explaining the influence of the degree of damage on prestressed CFRP-reinforced hollow RC box girders, using CFRP with over 30% prestress to reinforce hollow RC box girders with a 20% degree of damage poses a risk of CFRP–concrete interface debonding failure. Therefore, for hollow RC box girders with a 20% degree of damage, it is suitable to use 30% prestressed CFRP for reinforcement.

#### 4.2.3. CFRP–Concrete Interface

The damage to the CFRP–concrete interface in the hollow RC box girders reinforced with CFRP at different prestressing levels is shown in Figure 18. The figure shows that the interface damage increased with increases in prestressing levels, especially after the main steel bar yielded. High prestressing levels may not be an effective method for reinforcing the damaged girders, as it may lead to premature debonding of the CFRP–concrete interface. The figure reveals that the CFRP–concrete interface began to debond before the main bar yields in the case of 20% damaged specimens strengthened with a 40% prestressing level. The failure mode of the RC box girder reinforced by CFRP with high prestressing level changed from main bar yielding to CFRP–concrete interface debonding. A gradual increase in debonding length may cause the CFRP to detach from the anchoring end. Once the anchoring end fell off, the flexural resistance of the hollow box girder declined significantly. Therefore, while strengthening bridges with different degrees of damage, blindly increasing the prestressing level is not a suitable approach. It is more rational to select the CFRP with an appropriate prestressing level based on the degree of damage.

## 5. Conclusions

In this paper, the flexural behavior of damaged full-scale, hollow, reinforced concrete (RC) box girders reinforced with prestressed carbon fiber-reinforced polymer (CFRP) was studied through experiments and finite element analysis (FEA). The FE model incorporated the bond–slip relationship of the CFRP–concrete interface and considered debonding failure modes. The feasibility of the model was verified through experimental results. The influence of degree of damage and CFRP prestressing level on the flexural performance of damaged hollow box girders strengthened with prestressed CFRP was analyzed. Based on the analysis, the following conclusions were drawn:Prestressed CFRP significantly increases the yield and ultimate loads of damaged full-scale hollow RC box girders. Compared to unstrengthened RC girders, the yields and ultimate loads of RC girders reinforced with a 60% prestressing level increased by 25% and 33%, respectively.The degree of damage of the concrete hollow box girder greatly affects the strengthening effect. When the degree of damage is higher than 23%, the flexural bearing capacity of hollow RC box girders can only be restored to their original state after being strengthened with two CFRPs (50 mm width and 3 mm thickness) with a 30% prestressing level. However, if the degree of damage reaches 20%, the effect of a high prestressing level is not evident.When the degree of damage of the hollow RC box girder is higher than 35%, the failure mode of the strengthened hollow RC box girder is CFRP debonding. The failure mode of the hollow box girder changes from main reinforcement yielding to CFRP debonding if the degree of damage is 20% and the CFRP prestressing level is higher than 40%. Premature CFRP–concrete interfacial debonding failure is likely to occur when the CFRP prestressing level is high.Under the same load, the deflection of the hollow box girder is larger when the degree of damage of the girder is higher or the prestressing level of CFRP is smaller. However, the effect of degree of damage and prestressing level on the final deflection after reinforcement is minimal, as indicated by both experimental and FEA results.

## Figures and Tables

**Figure 1 materials-16-03338-f001:**
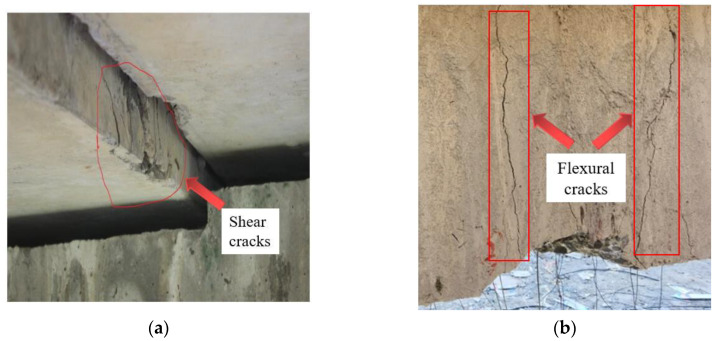
Typical damage to concrete hollow box girders: (**a**) shear cracks; (**b**) flexural damage; (**c**) parallel bending cracks; (**d**) roof damage; (**e**) steel bar corrosion.

**Figure 2 materials-16-03338-f002:**
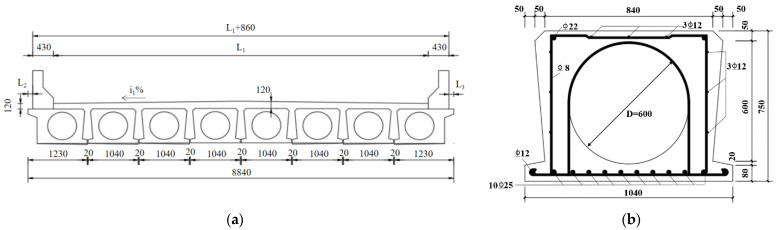
Detail of old bridge section: (**a**) the cross-section of the old bridge; (**b**) the cross-section of the box girder (all dimensions are in mm).

**Figure 3 materials-16-03338-f003:**
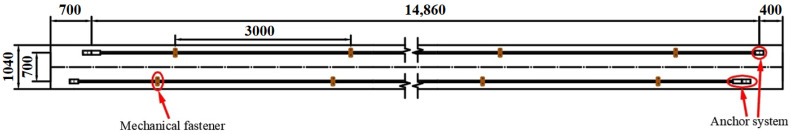
Prestressed CFRP on the bottom of damaged hollow box girder (all dimensions are in mm).

**Figure 4 materials-16-03338-f004:**
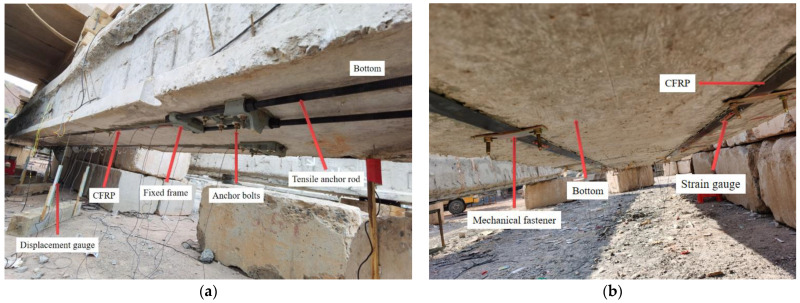
The site of anchor system: (**a**) anchor system; (**b**) prestressed CFRP.

**Figure 5 materials-16-03338-f005:**
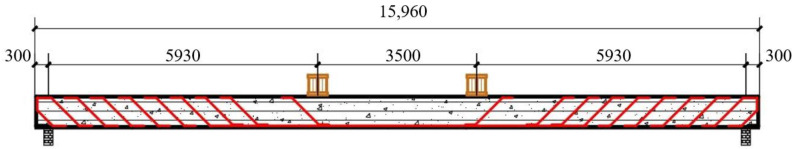
Four-point bending test (all dimensions are in mm).

**Figure 6 materials-16-03338-f006:**
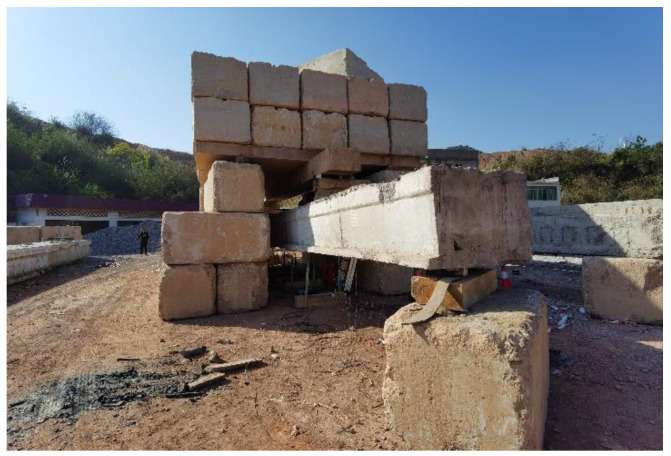
Test setup.

**Figure 7 materials-16-03338-f007:**
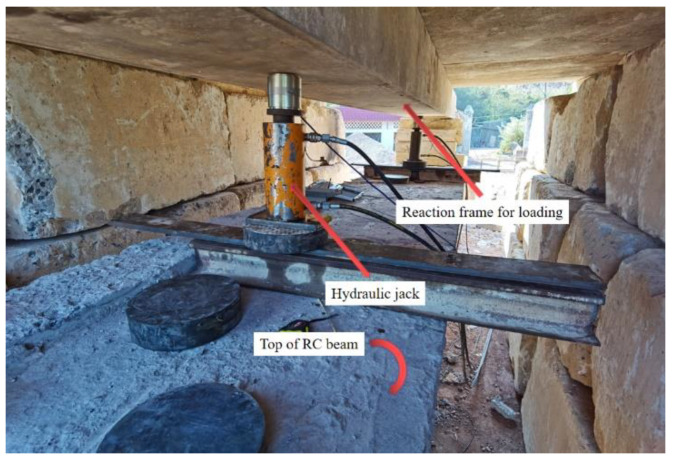
Loading device.

**Figure 8 materials-16-03338-f008:**
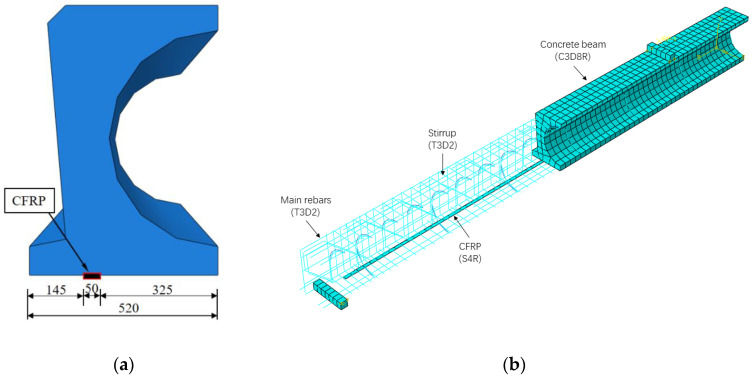
FE model of the prestressed CFRP-strengthened specimen: (**a**) half cross-section; (**b**) half of the hollow RC box girder.

**Figure 9 materials-16-03338-f009:**
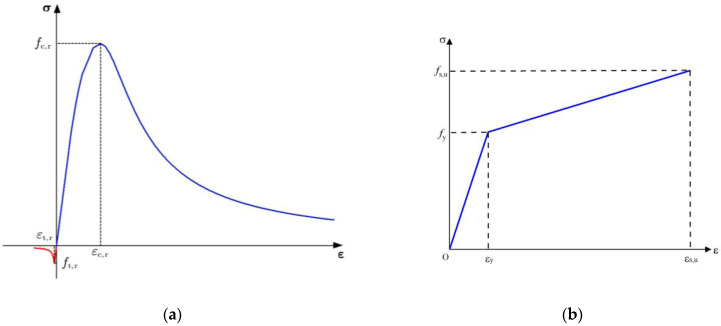
Stress–strain relationship of materials: (**a**) concrete; (**b**) steel reinforcement.

**Figure 10 materials-16-03338-f010:**
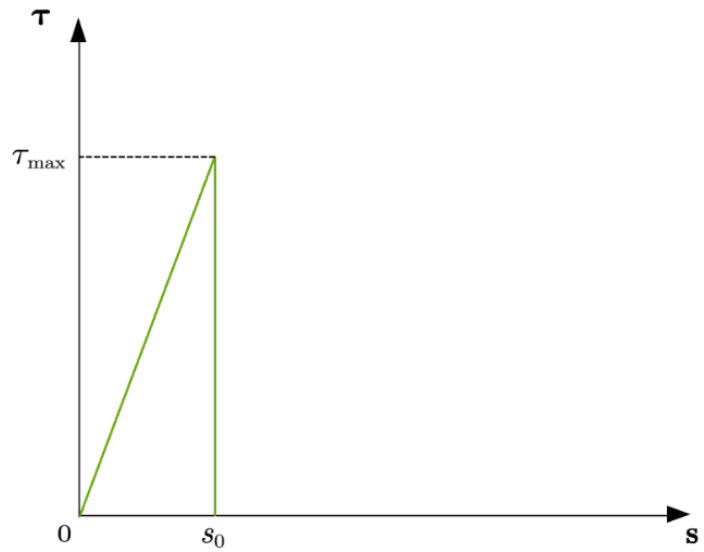
Bond–slip model [34].

**Figure 11 materials-16-03338-f011:**
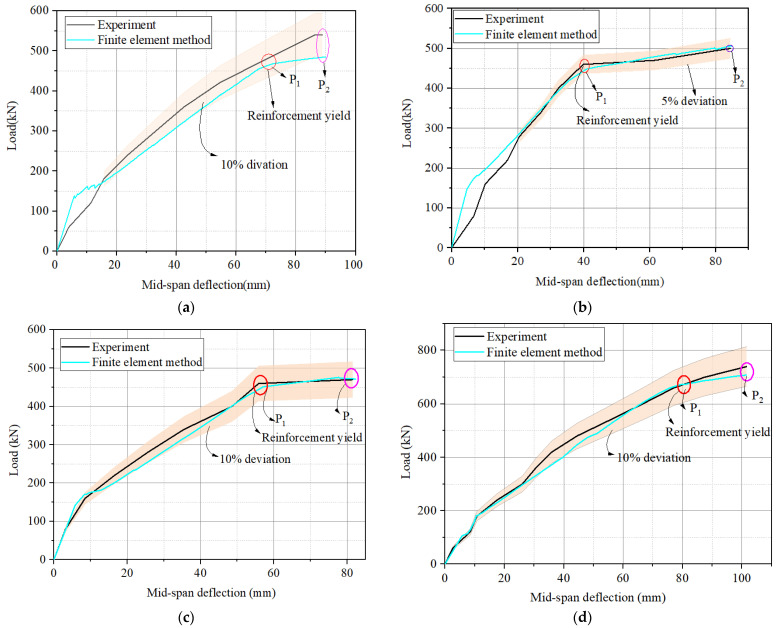
Comparison of experimental and finite element load–displacement curves: (**a**) G1, (**b**) G2, (**c**) G3, and (**d**) G4.

**Figure 12 materials-16-03338-f012:**
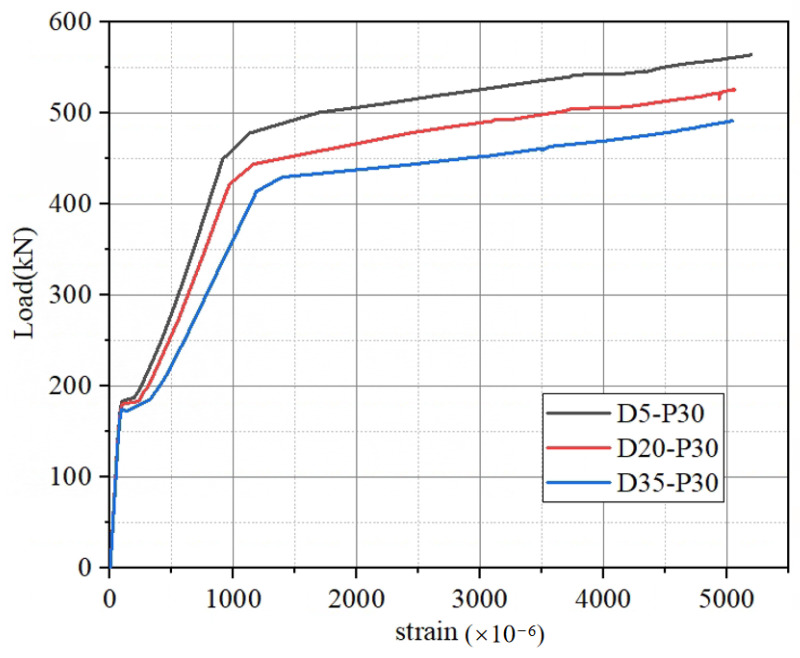
The load–strain curves of the main bars at different degrees of damage.

**Figure 13 materials-16-03338-f013:**
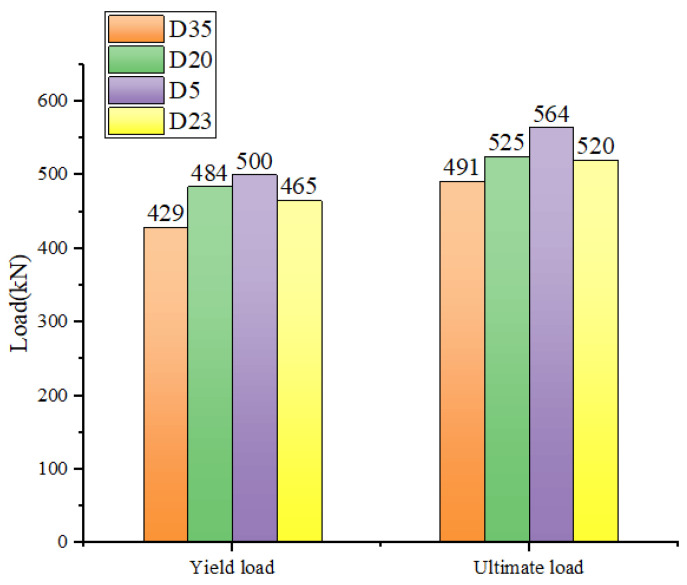
Comparison of maximum and yield loads of box girders with different degrees of damage.

**Figure 14 materials-16-03338-f014:**
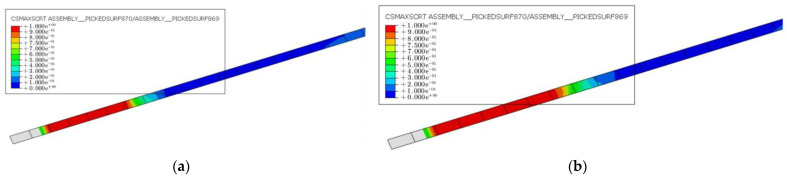
Shear stress distribution of CFRP–concrete interface at the ultimate load: (**a**) D5-P30, (**b**) D20-P30, and (**c**) D35-P30.

**Figure 15 materials-16-03338-f015:**
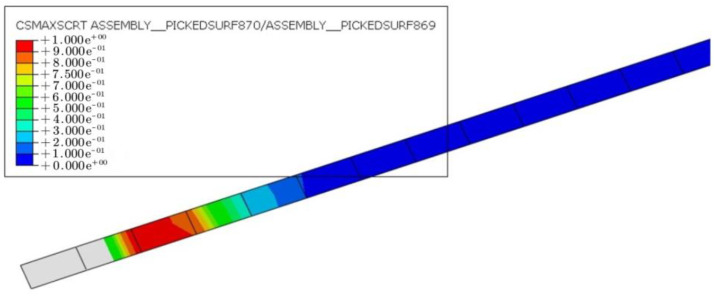
Shear stress distribution of CFRP–concrete interface of D35-P30 at yield load.

**Figure 16 materials-16-03338-f016:**
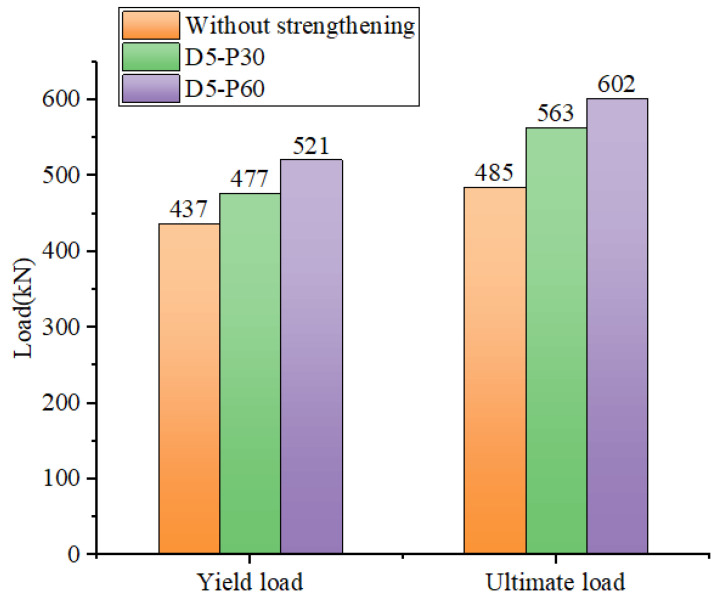
Comparison of the effect of different prestressing reinforcements in girders under 5% degree of damage.

**Figure 17 materials-16-03338-f017:**
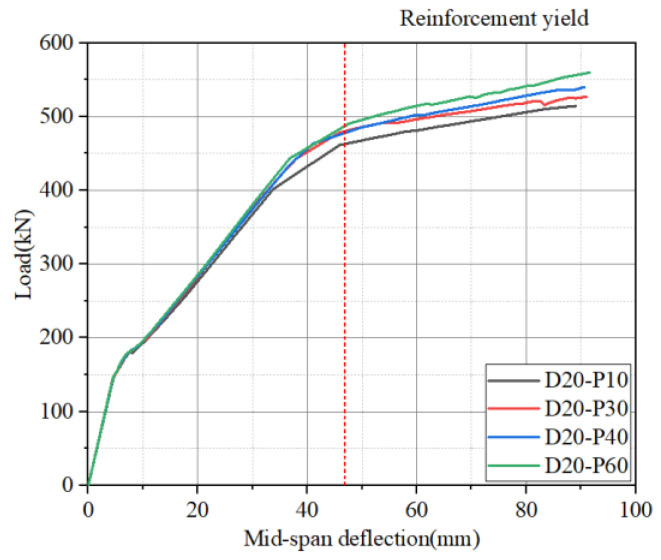
Load–deflection curves of hollow RC box girders with different prestressing levels at 20% degree of damage.

**Figure 18 materials-16-03338-f018:**
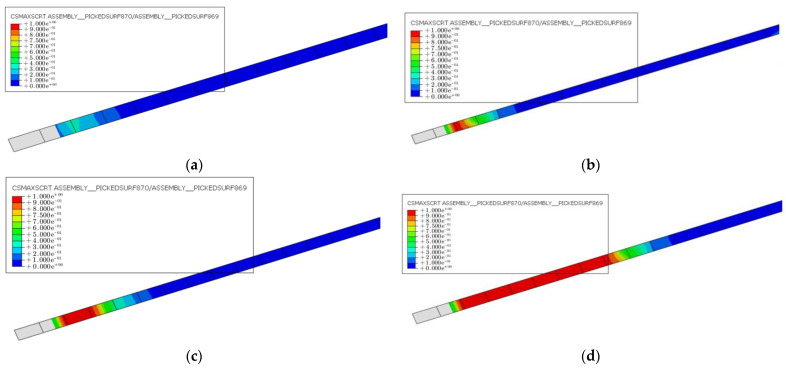
Damage cloud diagram of CFRP–concrete interface when the steel bars yield: (**a**) D20-P10, (**b**) D20-P30, (**c**) D20-P40, and (**d**) D20-P60.

**Table 1 materials-16-03338-t001:** Various parameters for different girders.

Girder	Damaged Degree	Strengthening Method	Yielding Load (kN)	Ultimate Load (kN)	Failure Mode
G1	7%	Without strengthening	480	540	SY, CP
G2	31%	30% prestressing	470	500	SY, CP, CC
G3	35%	40% prestressing	440	480	SY, PD, CC
G4	10%	60% prestressing	600	720	DB, PS, PF, CC

Annotations: SY = steel yielding, CP = lower flange concrete peeling, CC = concrete crushing, PD = CFRP partially debonding, DB = CFRP debonding failure, PS = CFRP sliding out of the anchor system, PF = CFRP fracture.

**Table 2 materials-16-03338-t002:** Yield and ultimate loads of beams D20-P30 at different mesh sizes.

Mesh Size of Concrete	Yield Load (kN)	Ultimate Load (kN)	Number of Meshes
75 mm	482.40	525.76	4664
100 mm	484.16	525.10	2320
200 mm	501.04	551.36	779

**Table 3 materials-16-03338-t003:** Comparison of finite element data corresponding to key points.

Load Corresponding Point Number	Corresponding Girder Number	Experimental Deflection/mm	Load		
			Calculated Value /kN	Experimental Value /kN	Relative Error/%
P1	G1	72.25	467.9	480	2.5
	G2	39.90	442.85	460	3.73
	G3	55.85	451.27	460	1.89
	G4	77.05	659.13	660	0.13
P2	G1	90.01	484.91	540	10.20
	G2	84.80	505.98	500	1.20
	G3	81.5	472.82	470	0.6
	G4	101.55	708.28	740	4.29

## Data Availability

Some or all data, models, or code that support the findings of this study are available from the corresponding author upon reasonable request.

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
