# Peer review of "Flexural Behavior of Damaged Hollow RC Box Girders Repaired with Prestressed CFRP"

_materials, 2023, doi:10.3390/ma16093338_

Round 1
Reviewer 1 Report
This paper introduces the flexural behavior of damaged hollow RC box girders repaired 2 with prestressed CFRP. This paper presents the flexural behavior of damaged full-scale hollow reinforced concrete 467 (RC) box girders reinforced with prestressed carbon fiber reinforced polymer (CFRP) was 468 studied through experiments and finite element analysis (FEA). The forthcoming research is suggested for publication after a major revision. The following items need to be explained:
1. The figures provided were not clear enough to make a decision. It is recommended to use better quality images. (For example, figure 11, 15, 18 and...)
2. Please, the authors add a summary of the relationship between the conducted research and past research at the end of the introduction. For example, a summary of the research conducted with general conclusions can be presented. In the following, more recent references should be used in the technical literature section.
3. A unique template should be used in the referencing section. Please revise all references.
4. Please indicate in all the figures and tables taken from the previous authors the official letter of permission to use or taken from them.
5. It is suggested to add a new figure that shows all the models schematically with details and materials used in the cross-section view.
6. Figure 1, cannot clearly show the damage caused, please use larger figures with higher quality.
7. In Table 1, state the reason for choosing 4 models with the mentioned features. What do the authors know about not choosing more models for a better final approximation?
8. Lines 413, 427 and... there are some typographical errors that can be seen in the entire article.
9. In the caption of Figure 12, the units of the horizontal graph should be given in SI units or equivalent explanations should be added to understand its meaning.
Reviewer 2 Report
The paper experimentally and numerically (by FE model) studied the flexural behavior of damaged full-scale hollow reinforced concrete (RC) box girders reinforced with prestressed carbon fiber reinforced polymer (CFRP). They investigated the capacity and shear stress distribution of FRP. The novelty is good. But, the revision is necessary.
1) Please add a notation list.
2) Fig. 11 needs more discussion in the text.
3) In order to provide a more comprehensive literature review, the authors should cite and discuss the following relevant papers in their revised manuscript:
- Static capacity of tubular X-joints reinforced with fiber reinforced polymer subjected to compressive load. Engineering Structures. 2021 Jun 1;236:112041.
- Local joint flexibility of tubular T/Y-joints retrofitted with GFRP under in-plane bending moment. Marine Structures, 2021;77, p.102936.
4) Please add the sensitive analyze on the mesh size in FE modeling.
5) In validation of the FE model, in fig. 11 a and 11b, why the initial stiffness of the FE model is considerably higher.
6) How could prestressed CFRP considerably rise the yield and ultimate loads?
Reviewer 3 Report
General Comments
The authors present an interesting work on the use of experimental and FEA methods to study the performance of prestressed CFRP as a tool for damage recovery. The study is justified and interesting conclusions are presented.
However, the manuscript contains some errors/shortcomings. In many instances, authors have made statements without citations to back the statements. Secondly, the authors often used the wrong words or inappropriate arrangement of words e.g. “damage degree” instead of “degree of damage”, etc.
These shortcomings and suggestions to remedy them have been made available to the authors under specific comments below.
Specific Comments
LINE 17: Write “experimental” instead of “experiment”.
LINE 23: Write “degree of damage” or “extent of damage” instead of “damage degree”.
LINE 28: Write “damage” instead of “damaged”.
LINE 63: References are needed immediately after this sentence.
LINE 81: Introduce the full meaning of the 3 acronyms and then their abbreviations in brackets immediately after each full name.
LINE 90: Introduce the full meaning of the 3 acronyms and then their abbreviations in brackets immediately after each full name.
LINE 112-113: Citation is needed to support this statement.
LINE 168: Delete “supply”.
LINE 179: The phrase “The strengthening proceeded include:” makes no sense. Rephrase to improve clarity and convey your thoughts without ambiguity.
LINE 182: Meaning is unclear. Please rephrase.
LINE 183: How was this external force applied? Indicate this.
LINE 213: Rephrase removing the phrase “this chapter analyzed”. There is no chapter(s) in your manuscript.
LINE 255: Citations are needed immediately after “Researchers have found”.
LINE 268: Citations are needed immediately after “previous studies”.
LINE 269: Citations are needed after the full-stop.
LINE 271: Write “values” instead of “results”.
LINE 281-282: Citations are needed to support this statement.
LINE 311: Write “load” instead of “loading”.
LINE 344: This line needs to be numbered (5.1.1) as a subheading or subsection.
LINE 349: Delete “growth”.
LINE 358: This line needs to be numbered (5.1.2) as a subheading or subsection.
LINE 361 and 362: Write “degree of damage” instead of “damage degree”.
LINE 363 and 364 and 367: Write “degree of damage” instead of “damage degree”.
LINE 370 and 371 and 374: Write “degree of damage” instead of “damage degree”.
LINE 376 and 379: Write “degree of damage” instead of “damage degree”.
LINE 384: This line needs to be numbered (5.1.3) as a subheading or subsection.
LINE 386: Write “below” instead of “under”.
LINE 394: Write “degree of damage” instead of “damage degree”.
LINE 413: This line needs to be numbered (5.2.1) as a subheading or subsection.
LINE 420: Write “results obtained” instead of “obtained results”.
LINE 424 and 427: Write “degree of damage” instead of “damage degree”.
LINE 428: This line needs to be numbered (5.2.2) as a subheading or subsection.
LINE 438, 440, and 442: Write “degree of damage” instead of “damage degree”.
LINE 449: This line needs to be numbered (5.2.3) as a subheading or subsection.
LINE 461 and 463: Write “degree of damage” instead of “damage degree”.
LINE 472: Write “degree of damage” instead of “damage degree”.
LINE 479 and 480 and 483: Write “degree of damage” instead of “damage degree”.
LINE 485 and 488: Write “degree of damage” instead of “damage degree”.
LINE 491 and 493: Write “degree of damage” instead of “damage degree”.
LINE 500-363: The numbering in square brackets at the beginning of each reference should be deleted.
Reviewer 4 Report
The article has studied the flexural behaviour of damaged hollow RC box girders repaired with prestressed CFRP. The study is interesting and important. However, several big issues should be well addressed before acceptance for publication. The reviewer’s comments are as follows:
(1) In Fig. 1, descriptions should be added around the dotted-line marks.
(2) Units must be added in Fig. 2, 3, 5, etc.
(3) More details should be compensated for the “damage level” or “damage degree”, how to calculate them?
(4) In Fig. 6, field condition is improper.
(5) In the modelling, important information is missing, e.g. ply sequence, mechanical properties, etc. for the CFRP. What is the type for CFRP? How to model the damage behaviour of CFRP? Have the authors conducted mesh-convergence analysis? It is recommended to review references, e.g., Int J Mech Sci 2019; 164: 105160, Int J lightweight mater 2023; 6: 108-116, Compos Struct 2017; 168: 322-334, Thin-Walled Struct 2023; 184: 110552, and add thorough information for numerical modelling.
(6) Number is missing for the equation in Section 3.4.
(7) Why the curve calculated by simulation initially fluctuates in Fig. 11(a)?
(8) In Section 5.1, etc., “Load-strain curve of main reinforcement” and so on must be numbered. Additionally, why it is load-strain curve instead of stress-strain curve?
(9) English must be carefully polished. e.g. “In this study, experiment and finite element analysis (FEA) was performed to investigate the flexural behavior of full-scale hollow RC box girders with varying degrees of damage strengthened by CFRP with different levels of prestress”, “If the damage degree of the hollow box girder was less than 23%, the flexural bearing capacity of the repaired girder could be recovered after being strengthened with two prestressed CFRP strips measuring 50 mm in width and 3 mm in thickness”, etc.
Round 2
Reviewer 1 Report
This is a well-written paper that discusses "Flexural behavior of damaged hollow RC box girders repaired with prestressed CFRP". The paper is properly written and well-structured and is acceptable to be published. And I think there are not any important comments to be addressed in the paper. (Please, authors only correct references in languages ​​other than English on page 12)
Author Response
Thanks. The references in languages ​​other than English on page 12 were deleted.
Reviewer 2 Report
The paper suggests for publication.
Author Response
Thanks.
Reviewer 4 Report
Several big issues have not been well addressed. For example, in Q4, the reviewer mean that the use of "field condition" is improper. What is filed condition? Q7 and Q8 are not addressed. Research is not performed by self-definition and self-sense.
